

# Predictive modeling of hourly probabilities for weather-related road accidents

Nico Becker[1,2], Henning W. Rust[1,2], and Uwe Ulbrich[1]

[1]Institut für Meteorologie, Freie Universität Berlin, Carl-Heinrich-Becker-Weg 6-10, 12165 Berlin, Germany
[2]Hans-Ertel-Centre for Weather Research, Berlin, Germany

**Correspondence:** Nico Becker (nico.becker@met.fu-berlin.de)

**Abstract.**

An impact of weather on road accidents has been identified in several studies with a focus mainly on monthly or daily accident counts. We study hourly probabilities of road accidents caused by adverse weather conditions in Germany on the spatial scale of administrative districts. Meteorological predictor variables from radar-based precipitation estimates, high-resolution reanalysis and weather forecasts are used in logistic regression models. Models taking into account temperature and hourly precipitation sums reach the best predictive skill according to different metrics. By introducing meteorological variables, the models hit rate is increased from 0.3 to 0.7, while keeping the false alarm rate constant at 0.2. Accident probability has a non-linear relationship with precipitation. Given an hourly precipitation sum of 1 mm, accident probabilities are about 5 times larger at negative temperatures compared to positive temperatures. Based on ensemble weather forecasts skilful predictions of accident probabilities of up to 21 hours are possible; the loss of skill compared to a model using radar and reanalysis data is negligible. The findings are relevant in the context of impact based warnings for both road users, road maintenance and traffic management authorities, as well as rescue forces.

## 1 Introduction

The road transport system is one of the most complex and dangerous systems that people have to deal with on a daily basis (Peden et al., 2004). In Germany, for example, road accidents lead to around 396,600 injuries and 3,200 fatalities per year in 2016(BASt, 2017). Causes for road accidents can be of technical, behavioral or environmental nature. Next to traffic volume, considered as the main cause for road accidents (e.g. Golob and Recker, 2003), weather is one of the most important factors contributing to road traffic safety. The impact of weather on road accidents has been addressed in several studies covering various temporal and spatial scales, focussing on different weather parameters and applying different methods; for a review see Theofilatos and Yannis (2014).

Two types of studies can be distinguished regarding the temporal scales. One type of study aims at relating road accidents to weather on a monthly or seasonal time scale (e.g. Fridstrøm et al., 1995; Shankar et al., 1995; Eisenberg, 2004; Bergel-Hayat and Depireb, 2004; Stipdonk and Berends, 2008). Aim of these studies is to gain insight into potential policy measures against the effects of adverse weather on road transport (Shankar et al., 1995). Due to the temporal variability of weather on





monthly time scales, such studies can only account for aggregate effects by considering for example the number of days with precipitation or the number of days with temperatures below the point of freezing (e.g. Fridstrøm et al., 1995). Other studies focus on daily timescales (e.g. Eisenberg, 2004; Keay and Simmonds, 2005; Caliendo et al., 2007; Brijs et al., 2008). On such time scales, the link between accident counts and the actual weather conditions on a specific day can be established. However,

the largest variability of traffic volume and accident rates is observed on sub-daily time scales, with peaks during the rush hours and low values during night time (Martin, 2002). Weather conditions may also change dramatically within hours. For taking into account the combined effect of weather and traffic volume, a sub-daily time scale is necessary. Nevertheless, only few studies focus on sub-daily time scales (e.g., Hermans et al., 2006a), possibly due to the lack of appropriate data sources. To establish robust relationships between accidents and weather parameters on an hourly time scale a sufficient amount of data is

required at a high spatial resolution. However, the analysis of highly resolved accident data is often subject to restrictions due to data protection directives. The spatial scales covered by the different studies vary from the national or state level (Hermans et al., 2006a) down to the level of individual cities (Yannis and Karlaftis, 2010) or specific roads or road segments (Ahmed et al., 2012).

Meteorological data used in accident studies is often derived from measurement stations. Either individual stations are used

(e.g. Knapp et al., 2000) or they are spatially aggregated for the area of interest (e.g. Eisenberg, 2004). In both cases, it might happen that not all relevant weather events are captured, because they do not hit a station. Recent studies use radar data to estimate the impact of precipitation on accidents (e.g. Mills et al., 2019). Jaroszweski and McNamara (2014) argue that radar data offers significant advantages over traditional station-based analyses, namely a better representation of rainfall due to a high spatial and temporal resolution.

Different weather parameters with a significant impact on road accidents have been identified. Depending on the study's modelling strategy and the specific formulation of variables characterising weather, magnitude and even the sign of the weather impact can vary between different studies. The most important weather parameter considered in most studies is precipitation. On wet roads the tire contact force is reduced (Hays, 2013), which increases the stopping distance starting at 100 km/h by about 20% compared to dry roads (Cho et al., 2007). Also glare caused by wet shining surfaces can lead to reduced visibility

and increase accident probabilities (Brodsky and Hakkert, 1988). Hermans et al. (2006a) study hourly crash counts in the Netherlands within a one-year period and found precipitation to be the most important factor among 17 different variables characterising weather.

On a monthly basis, snowfall can lead to reduced numbers of accidents, possibly due indirect effects like reduced traffic volumes or adaption of diving habits (Fridstrøm and Ingebrigtsen, 1991). On a daily basis, however, the direct effect of snowfall

was found to increase the accident risk. For example, Knapp et al. (2000) find that freeway crash rates increase by a factor of 13 in case of extreme snow storms. (Mills et al., 2019) find that injury and non-injury collisions increase by 66 and 137 percent, respectively, during winter storms. Winter storm events were identified using radar- and station-based observations. Malin et al. (2019) observe an sharp increase of relative accident risk if road surface temperatures drop below freezing point.

Since the first weather impact models for road accidents (Scott, 1986) various types of models have been used in this

context. Most popular are generalized linear models (GLMs), e.g. Poisson regression for accident counts or logistic regression



for accident probabilities (e.g. Fridstrøm et al., 1995; Caliendo et al., 2007; Keay and Simmonds, 2006), but also other methods like state-space (Hermans et al., 2006b) or autoregressive models (Brijs et al., 2008; Scott, 1986; Bergel-Hayat and Depireb, 2004) have been applied. Mostly, statistical models for weather impact on road accidents are used in an inferential way; they test hypotheses for variable relations by means of statistical hypothesis testing for parameters significance of prescribed

predictor variables, also referred to as explanatory modelling (Shmueli et al., 2010). This contrasts to predictive modeling, where statistical models are used for prediction of yet unobserved instances of the target variable (e.g., accident counts or probabilities). In practice, predictive models are built and assessed using cross-validation.

This study follows the predictive modeling approach: We build and asses the skill of logistic regression models for hourly probabilities of weather-related road accidents at the scale of administrative districts in Germany. Aim is to asses model per-

formance at small spatial and temporal scales, as well as identifying relevant meteorological predictor variables for optimizing the predictive skill. We thus seek an adequate functional relationship between hourly precipitation and accident probability under different temperature conditions and district characteristics. Instead of station-based observations, we use a gridded radar-based precipitation product and a new high-resolution reanalysis. Additionally, using ensemble weather forecasts, we assess the predictive skill of the accident model for leadtimes of up to 21 hours.

Section 2 describes data and preprocessing approaches. Statistical models and associated verification methods are described in Sect. 3. Results of model verification and the application of the models in a case study of a snowfall event are presented in Sect. 4, which is followed by a discussion and conclusions in Sect. 5.

## 2  Data

### 2.1  Accident data

A data set with anonymized information from police reports of all heavy road accidents in Germany from 2007 until 2012 is used (Source: Research Data Centre of the Federal Statistical Office and Statistical Offices of the Länder, *Statistik der Straßenverkehrsunfälle*, 2007-2012, own calculations). Heavy road accidents include all accidents with injuries, fatalities or write-offs. Minor accidents are not included in the data set. In total 4,313,069 complete accident reports are available for the period under investigation. Most accidents were indicated by the police as being caused by driver behaviour. However, almost

8% of the accidents were indicated as being caused by adverse road conditions, which includes a wet, snowy or icy road, but also mud or dirt on the road. This class of accidents, which we refer to as *weather-related accidents*, is selected to generate the response variable used in the logistic regression models. The location of the individual accidents is available on the level of administrative districts (*Landkreise*). Because of several territorial reforms during the study period, all accidents are assigned to boundaries of the 401 administrative districts as they existed in 2017. For each district an hourly time series is created for

the dichotomous variable *accident* being zero if no accident happened within the hour considered and one otherwise. In total this results in 16,775,572 data points, of which 136,559 contain at least one accident.



## 2.2 Radar-based precipitation data

Gridded hourly precipitation sums derived from the RADOLAN data set (Bartels et al., 2004) are available from the German Meteorological Service at a spatial resolution of $1 \times 1$ km . The RADOLAN combines radar reflectivities, measured by the 16 C-band Doppler radars of the German weather radar network, and ground-based precipitation gauge measurements. As

from radar reflectivities we cannot directly infer the precipitation amount at the ground but only the amount of reflection in the lower troposphere, observations from rain gauges are used to calibrate the precipitation amounts estimated from the radar reflectivities in an online-procedure typically used for nowcasting. Before calibration, a statistical clutter filtering is applied and orographic shadowing effects are corrected for. The RADOLAN projects thus aims at combining the benefits of high spatial resolution of the radar network with the accuracy of gauge-based measurements.

## 2.3 Reanalysis data

A reanalysis produced by a novel convective-scale regional reanalysis system for Central Europe (COSMO-REA2; Wahl et al., 2017) is used to generate meteorological predictor variables for the logistic regression models. The reanalysis results from the integration of COSMO-REA2 (a physical model for the atmosphere) with various heterogeneous observational data assimilated. COSMO-REA2 was developed within the framework of the Hans-Ertel Center for Weather Research (https://www.hans-ertel-zentrum.de). It contains different gridded atmospheric and surface variables for Central Europe at a spatial

resolution of 2 km and at hourly time steps. Deep convection is explicitly resolved by the model, while shallow convection is parameterized using the Tiedke scheme (Tiedtke, 1989). In addition to conventional station-based observations, radar-derived rain rates are assimilated using latent heat nudging. On hourly to daily time scales, the assimilation of radar information substantially improves the parameterized precipitation compared to other reanalysis datasets (Wahl et al., 2017).

## 2.4 Ensemble weather forecasts

Weather forecasts are used to study the predictability of accident probabilities based on weather forecasts with an ensemble prediction system (EPS). We use the regional high-resolution ensemble forecasting system COSMO-DE-EPS, which run operationally at the German Meteorological Service (DWD) before May 2018 with a spatial resolution of 2.8 km for the area of Germany. The COSMO-DE-EPS is initiated every 3 h with a leadtime $\tau$ of +21 h. $\tau$ is the difference between the time

the model is initialized and the time the forecast is valid for. For each initialization time 20 ensemble members are available, generated using different global model forecasts as initial and lateral boundary conditions and variations of parameterizations for unresolved processes as described in detail in Gebhardt et al. (2011) and Peralta et al. (2012). The spread of the ensemble members allows an estimation of the forecast uncertainty. Similar to COSMO-REA2, precipitation rates derived from radar observations are assimilated at forecast initialization using latent heat nudging (Stephan et al., 2008).

For our study, a post-processed product of the archived COSMO-DE-EPS forecasts for the years 2011 and 2017 was provided by the DWD. Instead of archiving the forecast data on the original model grid, area averages of $21 \times 21$ grid boxes ($56 \times 56 km$)





around 758 DWD owned gauge stations are stored. This drastically reduces the large amount of data, which facilitates their processing.

## 3 Methods

### 3.1 Data preparation

We aggregate the different meteorological variables to the level of administrative districts. For the station-based COSMO-DE-EPS forecasts a weighted mean of all available stations in the vicinity of the districts was calculated using the probability density function of a bi-variate circular symmetric normal distribution as the weighting function. A standard deviation of 25 km proved to be most appropriate, as it corresponds well to the average district area.

For a fair comparison of RADOLAN and COSMO-REA2 with COSMO-DE-EPS forecasts, the same aggregation is applied

to the gridded RADOLAN and COSMO-REA2 products: the areal averages around the 758 gauge stations is computed as described in section 2.4 and the data is aggregated to the district level by applying the weighting function, as described above.

### 3.2 Logistic regression

Logistic regression models are used to model the probability of a certain event based on independent predictor variables (e.g. Menard, 2002). Here, we model hourly accident probabilities. If $P_t$ is the probability that an accident occurs in a $1h$ time-

15 interval $(t - 1h, t]$, the logistic model equation is

$$P_t = 1/\{1 + \exp\left[-(\alpha + \mathbf{X_t}\,\boldsymbol{\beta})\right]\},\tag{1}$$

where $\alpha$ is the intercept term, $\mathbf{X_t} = (X_{t1}, ... X_{tn})$ the set of $n$ predictor variables, and $\boldsymbol{\beta} = (\beta_1, ..., \beta_n)$ are the corresponding parameters. $\alpha$ and the $\beta_i$ are estimated using maximum likelihood. If the effects of the two predictor variables $X_{ti}$ and $X_{tj}$ are not additive (i. e. the effect of $X_{ti}$ on $P_t$ depends on the state of $X_{tj}$), interaction terms can be added to the model equation. If

$X_{ti}$ and $X_{tj}$ are continuous variables, for example, this can be achieved by adding $\beta_{ij} X_{ti} X_{tj}$ to the linear term in Eq. 1, with $\beta_{ij}$ quantifying the combined effect of $X_{ti}$ and $X_{tj}$. For more detailed description of interactions, see Wood (2017).

The parameters of the logistic regression model can be easily converted to the odds ratio $\mathrm{OR} = \exp\beta_i$. The odds ratio for a given term $X_{ti}$ describes the change of the odds of the event to occur in case of a unit change in $X_{ti}$.

### 3.3 Assessing model performance

Parameter estimates $\hat{\beta}_i$ associated to individual predictor variables $X_{ti}$ can be tested for being significantly different from 0 using a two-tailed $z$-test (Dobson and Barnett, 2008).

Different logistic models are compared with information criteria. The most popular is the Akaike Information Criterion (AIC; Akaike, 1974) defined as

$$\mathrm{AIC} = 2k - 2\log(\hat{L}),\tag{2}$$




where $k$ is the number of parameters used in the model and $\hat{L}$ is value of the likelihood at its maximum. Fitted to the same data the model with lower AIC is to be preferred. The AIC penalizes models with more parameters to prevent overfitting.

The Brier Score (BS) is a proper score to measure accuracy of probabilistic forecasts for binary events, as they result from a logistic regression model. Based on Brier (1950) the BS can be defined as

$$\text{BS} = \frac{1}{N}\sum_{t=1}^{N}(f_t - o_t)^2\,, \tag{3}$$

where $f_t$ is the forecast probability, $o_t$ is the observed outcome of the event ($o_t \in \{0,1\}$), $t$ lables the events and $N$ is the total number of events. However, Benedetti (2010) has shown that the Brier Score may not be suitable when forecasting very rare (or very frequent) events. He suggests the use of the logarithmic score (or *absolute score*)

$$\text{LS} = a\frac{1}{N}\sum_{t=1}^{N}(o_t \ln f_t + (1-o_t)\ln(1-f_t))\,, \tag{4}$$

where $a = -(2\ln 2)^{-1}$ is simply a scaling factor, making LS comparable in size to BS. The LS is frequently used in the field of statistical mechanics and information theory and fulfills three basic desiderata (i.e. additivity, exclusive dependence on physical observations, and strictly proper behavior).

By defining a threshold $u$, a probabilistic forecast ($0 \leq f_t \leq 1$) can be transformed into a binary forecast, which is either *positive* (accident) or *negative* (no accident) if the forecast probability falls above ($f \geq u$) or below the threshold ($f < u$), respectively. The true positive rate (TPR, or hit rate) is the number of correctly predicted positive events divided by the total number of positive events. The false positive rate (FPR, or false alarm rate) is the number of incorrectly predicted negative events divided by the total number of negative events. The receiver operating characteristic (ROC) curve is a common way to illustrate the performance of a logistic regression model as a binary classifier, by plotting the TPR against the FPR for various thresholds $0 < u < 1$ (Hanley and McNeil, 1982). The area under the ROC curve (AUC) is frequently used for measuring the ability of a model to discriminate between positive and negative events. The AUC ranges between 0.5 and 1, which compares to random guessing and perfect discrimination, respectively. For a given FPR the corresponding TPR can be identified based on the ROC curve. In this study, we compare the TPR of different models while selecting $u$ so that the FPR is kept constant at 0.2.

A skill score $SS$ is a relative measure of how much a forecast $S_f$ outperforms a reference forecast $S_r$, defined as

$$SS = (S_f - S_r)(S_p - S_r)^{-1}\,, \tag{5}$$

where $S_p$ is the score of a perfect forecast. In this study we use the BS to compute the Brier Skill Score (BSS, $S_{p,BS} = 0$), the LS to compute the Logarithmic Skill Score (LSS, $S_{p,LS} = 0$) and the AUC to compute a skill score based on the ROC curve (AUCSS, $S_{p,AUC} = 1$).

While AIC penalizes large numbers of model parameters to avoid overfitting, in cross-validation techniques model parameter are estimated on a training data set and scores are computed on an independent testing data set. Here, we use a yearly cross-validation approach. Model parameters are estimated on a data set with one year of data left out and scores are calculated for





this respective year. This is repeated several times until for all years a score has been estimated. The score is then averaged over all years and used for model comparison.

To understand the behaviour of the model, the predicted accident probabilities of the regression models can be compared to non-parametric estimates for accident frequencies within bins of specific parameter ranges. For example, a predicted accident

probability for negative temperatures and a precipitation amount of 1 mm/h at 7:00 local time can be compared to the relative accident frequency for all time steps that showed negative temperatures and precipitation amounts in an interval of $1 \pm 0.1$ mm at 7:00. The uncertainty of model probability forecasts is estimated by computing the 95% confidence interval based on asymptotic standard errors. The uncertainty related to the non-parameteric estimates of the accident frequency is estimated by using a bootstrapping approach. The observed accident frequency is computed 10,000 times after drawing random samples with

replacement from the available data. The range between the 0.025 and 0.975 quantile of the resulting distribution of values can be used to construct a 95% confidence interval around the average observed accident frequency.

### 3.4    Model description

#### 3.4.1    Models without weather information

The models NULL and HOUR predict the accident probabilities for each district without using weather information (see Table 1

and Table 2 for a detailed description of predictor variables and models, respectively). The simplest model is the NULL model, using only the intercept and the time average accident probability $\overline{P}$ for each district as a predictor. $\overline{P}$ is transformed into $\overline{P}'$ using the inverse logistic function. By using $\overline{P}'$ in the logistic regression equation, a linear relationship between $\overline{P}$ and the hourly accident probability is established. By introducing $\overline{P}'$ we can distinguish between different districts using a single model parameter. Alternatively, we could include an individual intercept parameter for each district. However, this would require the

estimation of 401 parameters. By adding interaction terms the number of parameters would increase even more, making the model inapplicable.

The model HOUR includes an additional categorical variable $H$ specifying the time of day in hours (local time), which describes the diurnal cycle additionally to the average accident probability of each district. These two models are used as reference models to assess the benefit of adding weather information.

#### 3.4.2    Models using radar and reanalysis data

Accident, radar and reanalysis data overlap in time for the years from 2007 to 2012. For this time period, a binary predictor variable with hourly resolution for the near surface temperature $T_{REA}$ (temperature at 2 m height) is derived from COSMO-REA2, which distinguishes between temperatures above and below 0°C. Furthermore, a continuous variable $Pr_{RAD}$ with the hourly precipitation sum in mm/h is used. In model RAD the model HOUR is extended by adding $T_{REA}$ and $(Pr_{RAD})^{0.2}$

as direct effects. Different combinations of exponents have been tested to transform the precipitation, but 0.2 lead to the best results in terms of model skill. In the model RAD_INT the two-point interaction terms between $\overline{P}'$, $H$, $T_{REA}$ and $(Pr_{RAD})^{0.2}$ are added to the model equation. Model parameter estimates result from using data from all districts simultaneously. However,




the skill scores are calculated for each district individually within the cross validation procedure. This allows to asses the performance of the model for different districts.

Additionally, we fit the models to the individual districts, yielding models RAD_IND and RAD_INT_IND[1], respectively. On the one hand, these models capture the district specific characteristics; on the other hand, the amount of available data points

for each model is strongly reduced, which complicates the estimation of model parameters, in particular for districts with low accident numbers. These models are used to quantify the benefit of having one model for all districts.

### 3.4.3  Models using weather forecast data

The overlapping time period of accident data and COSMO-DE-EPS data are the years 2011 and 2012. For this time period temperature and precipitation is aggregated to district level as before for all 20 ensemble members. This is done separately for

all forecast leadtimes $\tau$, ranging between 1 h after forecast initialization and 21 h after initialization.

The COSMO-DE-EPS provides hourly forecast data, but is initialized only every three hours. Therefore, not all hours are available for all leadtimes. E.g. a leadtime of 6 h is only available at 0, 3, 6, 9, 12, 15, 18, 21 UTC, while a leadtime of 7 h is only available at 1, 4, 7, 10, 13, 16, 19, and 22 UTC. Furthermore, the logistics regression model uses local time, which has to take into account daylight savings time. Both effects complicate an explicit use of the hour as a predictor variable in

combination with COSMO-DE-EPS data. Therefore, to facilitate the incorporation of a diurnal cycle in the model, a two step procedure is applied. First, the model HOUR is used to forecast the average diurnal cycle of accident probabilities $P_H$ for each district. Then $P_H$ is transformed into $P'_H$ using the inverse logistic function. Second, $P'_H$ is used to replace the terms $\overline{P}' + H$ (compare HOUR and EPS_HOUR in Table 2, for example).

Three different ways to incorporate the ensemble information in the models are used.

1. *Deterministic forecasts:* In case of the model EPS_MEM$_i$_INT an individual set of parameters is estimated for each ensemble member and each leadtime. Skill scores are calculated for each of the resulting sets of parameters separately, thus treating the ensemble members as single deterministic forecasts.

   2. *Meteorology-averaged ensemble:* In case of the model EPS_MEAN_INT the parameters are estimated using the ensemble mean of the meteorological variables, which results in a single set of parameters for each leadtime.

3. *Probability-averaged ensemble:* In case of the model EPS_PMEAN_INT accident probabilities are predicted using the models EPS_MEM$_i$_INT for the individual ensemble members, but the ensemble mean of the predicted probabilities is calculated before using it to compute the scores in the cross-validation procedure.

The models EPS_HOUR, EPS_RAD_INT correspond to the models HOUR, RAD_INT, but are fitted separately to the data available for each leadtime time, to allow a direct comparison to the models using COSMO-DE-EPS data.

---

[1]INT refers to the use of interaction terms in the model equation, while IND refers to estimating model parameters for each district individually.



## 4 Results

### 4.1 Models using radar and reanalysis data

The time average hourly probability that at least one weather-related accident occurs in an administrative district is referred to as $\overline{P}$. It ranges from below 0.001 for smaller districts with less inhabitants to more than 0.05 for densely populated cities. The NULL model simply gives $\overline{P}$ for each district and serves as a reference model. As expected, the AUC is 0.5, indicating that the model is not able to distinguish between accident and non-accident cases (Table 3).

In model HOUR all parameters of the categorical variables $H$ are significantly different from zero with p-values below 0.001, indicating that the diurnal cycle is an important aspect of the accident characteristics. The average AUC of all districts is 0.62, indicating that the introduction of the hour as a predictor improves the model.

The introduction of temperature and precipitation as direct effects in the model RAD leads to a further improvement of the scores, compared to NULL and HOUR. With an AUC of about 0.81 and an AUCSS of 0.49 (HOUR as reference) temperature and precipitation can be considered useful in terms of binary classification of accident events. The TPR increases from 0.3 for HOUR to 0.7 for RAD. The interaction terms in RAD_INT slightly improve all scores except for the TPR.

Fig. 1 shows that the variability of the AUCSS values of the different districts is relatively large, compared to the differences between the models. However, there is no evident systematic relationship between the skill of the model and the geographic location of the district or the district specific topography (not shown).

Fig. 2 shows the modelled accident probabilities (solid lines) predicted by the RAD (left) and RAD_INT (right) versus precipitation (top), hour (middle) and $\overline{P}$ (bottom) together with the 95% confidence intervals estimated from the standard errors (shaded). Additionally, the accident probabilities estimated non-parametrically (number of time steps with accidents divided by total number of time steps) are shown (markers) together with the 95% confidence intervals estimated using a bootstrapping approach (vertical lines). Model and non-parametric probabilities are shown for positive (red) and negative (blue) temperatures.

The modelled accident probabilities as function of $Pr_{RAD}$ are shown for 7:00 local time for a district with an average probability for weather-related accidents of $\overline{P} = 0.01$ (Fig. 2, top row). Non-parametric probability estimates are calculated for precipitation bins with a width of 0.1 mm/h including only districts with $\overline{P} = 0.01 \pm 0.002$. In general, accident probabilities are lowest at $Pr_{RAD} = 0$, show a steep increase with increasing precipitation with a decreasing slope at higher precipitation rates. Probabilities are higher at temperatures below 0°C. At $Pr_{RAD} 1\ mm/h$ they are about 5 times higher if temperatures are below 0°C. For RAD the modelled probabilities fit well to the non-parametric probability estimates at $Pr_{RAD} < 0.5$ mm/h, but overestimate probabilities at higher precipitation rates. In contrast, the model RAD_INT shows reduced probabilities, which fit much better to the non-parametric probability estimates. The curved shape of the functional relationship between precipitation and probability is realized by taking precipitation to the power of 0.2. The value 0.2 was found to be the best choice after testing a series of different exponents, other functional relationships as $\log 1 + Pr$, as well as categories of precipitation.

The modelled probabilities as function of $H$ are shown for $Pr_{RAD} = 0$ mm/h (solid lines) and $Pr_{RAD} = 0.5$ mm/h (dashed lines) for $\overline{P} = 0.01$ (Fig. 2, middle row). Non-parametric probability estimates are calculated using time steps with $Pr_{RAD} = 0$ mm/h (circles) and $Pr_{RAD} = 0.5 \pm 0.25$ mm/h (triangles) including only districts with $\overline{P} = 0.01 \pm 0.002$. In general, accident



probabilities show a pronounced diurnal cycle with maximum probabilities during morning and afternoon rush hours. RAD overestimates the observed probabilities in particular during the morning hours with precipitation at negative temperatures. The model RAD_INT is able to capture the observed diurnal cycle more precisely.

The modelled probabilities as function of $\overline{P}$ are shown for $Pr_{RAD} = 0$ mm/h (solid lines) and $Pr_{RAD} = 0.5$ mm/h (dashed

lines) at $H = 7$ h (Fig. 2, bottom row). Non-parametric probability estimates are calculated using time steps with $Pr_{RAD} = 0$ mm/h (circles) and $Pr_{RAD} = 0.5 \pm 0.25$ mm/h (triangles) including districts with $\overline{P} = 0.01 \pm 0.002$. In general, the probabilities show an monotonic increase with $\overline{P}$, which justifies the introduction of $\overline{P}$ as a predictor to distinguish between different districts. The predictions of RAD and RAD_INT are relatively similar and lie mostly within the confidence intervals of the observed probabilities.

In a next step, we compare the models RAD and RAD_INT, which are fitted to all districts simultaneously, to the models RAD_IND and RAD_INT_IND, which are fitted to all districts individually. Fig. 3 shows the difference of the AUCSS between RAD and RAD_IND (red) and between RAD_INT and RAD_INT_IND (black) as a function of $\overline{P}$. $\overline{P}$ provides a direct information about how many accident cases were available in the time series used for training the models. In general, the AUCSS differences are mostly negative, indicating that the models fitted to each district individually perform poorer than the models

including all districts. The AUCSS differences decrease with increasing $\overline{P}$, i.e. increasing accident numbers. Furthermore, the AUCSS differences are larger for the more complex models with interaction terms. The results are similar for the LSS (not shown).

Based on the results of this section, we can conclude that RAD_INT should be preferred over RAD, since it achieves the best scores and better represents the functional relationship between probability and precipitation as well as the diurnal cycle.

Furthermore, RAD_INT preforms better than RAD_INT_IND, which is fitted to each district individually.

## 4.2 Models using weather forecast data

The model RAD_INT showed the best performance among the models predicting accident probability using radar and reanalysis data (Sect. 4.1). In this section the model formulation of RAD_INT is modified to allow the use of COSMO-DE-EPS ensemble weather forecasts. To facilitate the modelling procedure, the variables $H$ and $\overline{P}$ are combined into a single vari-

25 able $P'_H$ by using the model HOUR, which effectively results in a district-specific diurnal cycle of accident probabilities (see Sect. 3.4.3 for details). $P'_H$, precipitation and temperature are used as predictor variables, including their interaction terms. In case of all of the following models, a new set of parameters is estimated for each leadtime from 1 to 21 h, using only those time steps, which are available for the specific leadtime.

The model EPS_RAD_INT uses $T_{REA}$ and $Pr_{RAD}$ and serves as an upper limit, representing the best available model

based on reanalysis and radar data. The AUCSS of EPS_RAD_INT as a function of leadtime shows a repetitive pattern with maximum values of around 0.5 at leadtimes 1, 4, 7, etc. and 0.47 in between (Fig. 4, orange line). This repetitive pattern can be explained by different data time steps that go into the model training and verification at the different leadtimes due to the three-hourly initialization of the COSMO-DE-EPS.





The model EPS_MEMi_INT is estimated for each of the 20 ensemble members individually, which therefore results in 20 deterministic forecasts with 20 individual AUCSS values per leadtime. The AUCSS drops from 0.48 at leadtime 1 h to below 0.45 at leadtime 21 h (gray lines). The spread between the AUCSS of the different ensemble members increases with increasing leadtime. The model EPS_MEAN_INT is based on the ensemble mean of the meteorological variables (meteorology-

averaged ensemble) and shows a slightly higher AUCSS (black solid line) than all the deterministic forecasts. The model EPS_PMEAN_INT, which is based on the ensemble mean of the accident probabilities of the 20 versions of EPS_MEMi_INT (probability-averaged ensemble), shows again a slightly higher AUCSS (black dashed line) than the meteorology-averaged ensemble. As expected, the AUCSS values of all models based on weather forecast data are lower than the AUCSS of EPS_RAD_INT based on radar and reanalysis data. However, the differences are relatively small. The LSS shows a simi-

lar behaviour regarding the leadtime dependence as the AUCSS (not shown).

## 4.3  Case study

The models RAD_INT and EPS_PMEAN_INT are used in a case study with adverse winter weather conditions on Dec. 3rd, 2012. At temperatures below the freezing point the fronts of a low pressure system lead to snowfall in large parts of Germany. These weather conditions lead to a total number of 280 accidents caused by road condition. The majority of the accidents

occurred in southern and western Germany[2].

For the district of Stuttgart, which was located within the affected area, the RADOLAN data shows low precipitation amounts in the early morning and higher precipitation amounts of up to 0.3 mm/h in the afternoon (Fig. 5a). The COSMO-DE-EPS forecast, initialized on Dec. 3rd, 2012 at 00:00 UTC (02:00 h local time), shows ensemble mean precipitation amounts of more than 0.6 mm/h in the afternoon and a large spread between the ensemble members.

The temperature in COSMO-REA2 is below 0°C until 19:00 h and then changes to warmer conditions (Fig. 5b). All ensemble members of COSMO-DE-EPS predict the change to positive temperatures two hours earlier than observed.

The accident probability of EPS_RAD_INT shows the combined effect of the average diurnal cycle, RADOLAN precipitation and COSMO-REA2 temperature (Fig. 5c). It shows a peak of 0.07 in the morning during rush hour at low precipitation amounts at freezing temperatures, a drop to 0.02 at noon when RADOLAN shows no precipitation, a maximum peak of 0.22

in the afternoon, when precipitation is strongest. In general, the accident probability of EPS_PMEAN_INT matches well with EPS_RAD_INT. However, it slightly overestimates the morning peak and overestimates the afternoon peak due to the too intense and persistent precipitation.

The hourly accident probability $P$ is useful for authorities to assess how likely the occurence of an accident is in a certain district at a certain point in time. However, it does not reflect the risk of an individual road user, as it does not distinguish

whether $P$ changes due to weather-related effects, due to a change in traffic density along the diurnal cycle, or due to the district characteristic. For example, a road user travelling from a district with a high average accident probability $\overline{P}$ to a district with a low $\overline{P}$ would observe a decrease of $P$, also if the weather conditions remain the same. Therefore, to estimate the

---

[2]Due to regulations regarding anonymization and data protection we are not allowed to show accident counts less than three, which prevents us from showing accident counts for single hours or days at the district level.



impact on an individual road user, we compare $P$ to $P_0$, the probability under conditions without precipitation and positive temperatures (Fig. 5c, dotted line). The fraction $P/P_0$ gives the amplification of the actual predicted probability $P$ compared to warm and dry conditions (Fig. 5d). In case of the forecast for Dec. 3rd, 2012, the amplification factor ranges between 50 in the afternoon when the precipitation amount is high and 5 around noon when precipitation amount is low. This factor could be

a potential weather impact forecast product.

On Dec. 3rd, 2012 at 17:00 h local time, the COSMO-DE-EPS overestimates the precipitation amount in large parts of western and southern Germany, compared to RADOLAN (Fig. 6). The area with temperatures below 0°C is captured relatively well, compared to COSMO-REA2. The accident probability $P$ is largest where high precipitation amounts and freezing temperatures occur. Spatially, $P$ is relatively inhomogeneous, which reflects the large differences in average accident probability

between the individual districts. $P/P_0$, representing the increase in accident probability of individual drivers, is spatially more homogeneous.

## 5   Summary, discussion and conclusions

Police reports of heavy road accidents in Germany were used to construct hourly time series based on weather-related accidents caused by adverse road conditions for German administrative districts. Different meteorological datasets aggregated to district

level were used in logistic regression models to predict hourly accident probabilities. Models of different complexity were compared after calculating different skill scores using a yearly cross-validation approach. The best model with respect to these scores included district-specific average accident probability, the hour of the day, hourly precipitation and temperature, as well as their interaction terms. By introducing meteorological variables to the model, the hit rate (TPR) could be increased from 0.3 to 0.7, while the false alarm rate (FPR) was kept constant at 0.2. It was shown that the probability of weather-related accidents

increases non-linearly with increasing hourly precipitation. Given an hourly precipitation of 1 mm, the accident probability is approximately 5 times higher at negative temperatures, compared to positive temperatures. In a case study it was shown that the model is able to reasonably capture the spatial and temporal development of accident probabilities during adverse winter weather conditions. When using ensemble weather forecasts to predict accident probabilities, the skill of the logistic regression model remains almost constant for a forecast leadtime of up to 21 h. Furthermore, the use of ensemble forecasts leads to a

higher skill compared to a setting, where ensemble members are treated as individual deterministic forecasts. These findings are in line with the results of Pardowitz et al. (2016), who show that the use of ensemble information improves predictions of storm damage probabilities.

It is known that the main parameters affecting accident probability are traffic flow and density. In an optimal case one would used measurements of these variables as a model predictor for accident probability. However, traffic measurements are not continuously available for all administrative districts. Additionally, measurements of traffic flow are mainly available

for highways and federal roads and might not be representative for municipal roads, where the majority of the accidents occur. Furthermore, in an operational setting, where the model is applied for predicting future accident probabilities, traffic measurements are not available. Therefore, we decided not to directly include traffic measurements in the models. Instead, the





hour of the day was used as a categorical predictor variable to capture the average diurnal cycle of accident probability. It was shown, that this approach is able to reasonably represent the inner-day variability of accident probability. The introduction of additional factors like a weekends or holidays did not lead to a significant improvement of the model.

5 It is a challenging task to combine accident data, which is available for the area of administrative districts, with meteorological data, which is usually available in the form of point observations or gridded data. Different ways of aggregating meteorological data to district level were tested and the approach based on distance-weighted averaging, which is presented in this study, showed the best results.

The temperature at 2 m height was used in this study to include the effect of negative temperatures in the statistical model in a relatively simple approach. It has the benefit, that the temperature at 2 m height is a well established meteorological 10 parameter, which is measured at most stations and available in all weather forecasting models. However, it might not reflect the conditions at the road surface, which can deviate from the conditions at 2 m height. Also the choice of 0°C as a fixed threshold is a simplified approach, since ground frost or snowfall could also occur at higher 2 m temperatures. By using non-linear approaches like generalized additive models (Wood, 2017) a smooth transition between positive and negative temperatures could be established in future studies. Furthermore, it might be detrimental that area averaged temperatures are used, which 15 does not fully represent topographic variations within the area of a certain district. A more complex approach could make use of a road surface model, which includes the combined effects of precipitation, evaporation and road surface temperatures in a more sophisticated way (e.g. Juga et al., 2013).

In addition to the weather parameters presented in this study, other parameters like snow fall amount or combined measures of cloud cover and sun angle to describe the impact of sun glare were tested as potential predictor variables. Furthermore, 20 advanced predictor selection techniques like genetic algorithms (Calcagno et al., 2010) and the least absolute shrinkage and selection operator (Tibshirani, 1996) were applied, to find optimal combinations of parameters. However, none of the results were able to significantly improve the skill of the best models presented in this study, as measured by the cross-validation approach.

We found that the probability of weather-related accidents increases approximately hourly precipitation to the power of 0.2. 25 This should not be understood as a universal relationship. Instead, it is likely to depend on different aspects of road system (e.g. how fast is the water able to leave the road surface) or the average car characteristics (e.g. the share of cars equipped with assistance systems, or the type tires). It may even change in time, as road and car qualities improve.

In this work we showed two ways of modelling probabilities in different districts: first, by creating a model that distinguishes between different districts based on their average accident probability and, second, by creating a model for each district in- 30 dividually. We found that to first approach lead to higher skill scores, particularly for districts with low accident numbers. Including additional district-specific parameters describing the characteristics of the road network or topographic conditions could help to further refine the model.

This study shows that skillful relationship between meteorological parameters and weather-related road accidents can be established. Forecasts of probabilities of weather-related road accidents, as presented in this study, might be useful for authorities 35 (traffic management, police or emergency services) on the one hand and road users on the other hand. However, it is reasonable





to provide the information about accident risk in different, user specific formats, which were introduced in Sect. 4.3. Authorities might be primarily interested in aggregated risk information for their region of interest, e.g. the occurrence probability of accidents in an administrative district. On the other hand, a road used is rather interested in his individual risk. The individual risk is better reflected through an amplification of risk compared to certain reference conditions (e.g. warm and dry weather).

It was shown that impact-based warning can lead to a better actions of the recipients (Weyrich et al., 2018). Furthermore, Hemingway and Robbins (2019) state that information about weather impacts can be helpful for operational meteorologists when issuing weather warnings. This was found using a prototype impact model for predicting the risk of road disruption due to wind-induced overturning of vehicles. In this context, the accident model presented in this study can be considered a useful tool for reduction of road traffic risk.

*Data availability.* The accident data for Germany was obtained from the Research Data Centre of the Federal Statistical Office and Statistical Offices of the Länder. The RADOLAN and COSMO-REA2 data are publicly available from the Climate Data Center of the Deutscher Wetterdienst. The COSMO-DE-EPS data was provided by the Deutscher Wetterdienst upon request.

*Author contributions.* Data analysis and visualization was done by NB; All authors contributed to writing the manuskript.

*Competing interests.* The authors declare that they have no conflict of interest.

*Acknowledgements.* This research was carried out in the Hans-Ertel-Centre for Weather Research. This research network of universities, research institutes, and the Deutscher Wetterdienst is funded by the BMVI (Federal Ministry of Transport and Digital Infrastructures).



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



**Table 1.** Descriptions of predictor variables used in different logistic regression models for hourly probabilities of weather-related road accidents in Germany administrative districts.

| Name | Description |
| --- | --- |
| $\overline{P}$ | Temporal average of accident probability of in an administrative district |
| $\overline{P}' = -log(1/\overline{P}) - 1$ | $\overline{P}$ transformed using the inverse logistic function. |
| $H$ | A categorical variable for the hour of the day |
| $Pr_{RAD}$ | Hourly precipitation in mm from RADOLAN data aggregated to district level |
| $Pr_{EPS,i}$ | Hourly precipitation in mm from i[th] ensemble member of COSMO-DE-EPS aggregated to district level |
| $Pr_{EPS,m}$ | Ensemble mean of hourly precipitation in mm calculated from COSMO-DE-EPS ensemble members aggregated to district level |
| $T_{REA}$ | A binary variable indicating whether the COSMO-REA2 near surface temperature aggregated to district level is above or below 0°C |
| $T_{EPS,i}$ | As $T_{REA}$ but derived from the i[th] ensemble member of COSMO-DE-EPS |
| $T_{EPS,m}$ | As $T_{REA}$ but derived from the ensemble mean of COSMO-DE-EPS |
| $P_H$ | Accident probability as predicted by model HOUR (see Table 2) based on $\overline{P}$ and $H$ |
| $P_H' = -log(1/P_H) - 1$ | $P_H$ transformed by the inverse logistic function |




**Table 2.** Description of different logistic regression models for hourly probabilities of weather-related road accidents in Germany administrative districts and their degrees of freedom (Df). Formulas are written using the statistical formula notation system as used in programming languages as R and Python, with colons indicating interaction terms.

| Name | Formula | Df |
|------|---------|-----|
| *models using radar and reanalysis data (2007-2012)* | | |
| NULL | $y \sim 1 + \overline{P}'$ | 2 |
| HOUR | $y \sim 1 + \overline{P}' + H$ | 25 |
| RAD | $y \sim 1 + \overline{P}' + H + T_{REA} + (Pr_{RAD})^{0.2}$ | 27 |
| RAD_INT | $y \sim 1 + \overline{P}' + H + T_{REA} + (Pr_{RAD})^{0.2} + P:H + P:T_{REA} + P:(Pr_{RAD})^{0.2} + H:$ $T_{REA} + H:(Pr_{RAD})^{0.2} + T_{REA}:(Pr_{RAD})^{0.2}$ | 99 |
| RAD_IND | As RAD but without $\overline{P}'$, fitted to all 401 districts individually | $401 \times 29$ |
| RAD_INT_IND | As RAD_INT but without $\overline{P}'$, fitted to all 401 districts individually | $401 \times 73$ |
| *models using radar, reanalysis and weather forecast data (2011-2012)* | | |
| EPS_HOUR | $y \sim 1 + P'_H$ | 2 |
| EPS_RAD_INT | $y \sim 1 + P'_H + T_{REA} + (Pr_{RAD})^{0.2} + P'_H:T_{REA} + P'_H:(Pr_{RAD})^{0.2} + T_{REA}:$ $(Pr_{RAD})^{0.2}$ | 6 |
| EPS_MEM$_i$_INT | $y \sim 1 + P'_H + T_{EPS,i} + (Pr_{EPS,i})^{0.2} + P'_H:T_{EPS,i} + P'_H:(Pr_{EPS,i})^{0.2} + T_{EPS,i}:$ $(Pr_{EPS,i})^{0.2}$ | 6 |
| EPS_MEAN_INT | $y \sim 1 + P'_H + T_{EPS,m} + (Pr_{EPS,m})^{0.2} + P'_H:T_{EPS,m} + P'_H:(Pr_{EPS,m})^{0.2} +$ $T_{EPS,m}:(Pr_{EPS,m})^{0.2}$ | 6 |
| EPS_PMEAN_INT | As EPS_MEM$_i$_CON_INT, but using ensemble mean probabilities for verification | $20 \times 6$ |





**Table 3.** Verification measures for models using radar and reanalysis data (2007-2012). Scores computed in a yearly cross-validation approach for each administrative district are shown as averages of all districts. The best value of each score is underlined.

| Model | NULL | HOUR | RAD | RAD_INT | RAD_IND | RAD_INT_IND |
|-------|------|------|-----|---------|---------|-------------|
| AIC | 1885238 | 1856974 | 1629688 | 1624719 | - | - |
| AUC | 0.5000 | 0.6157 | 0.8056 | 0.8097 | 0.7977 | 0.7740 |
| TPR | - | 0.3252 | 0.6715 | 0.6707 | 0.6644 | 0.6366 |
| LS | 0.0324 | 0.0319 | 0.0280 | 0.0279 | 0.0282 | 0.0302 |
| BS | 0.0079 | 0.0079 | 0.0077 | 0.0077 | 0.0077 | 0.0077 |
| AUCSS | -0.3053 | 0.0000 | 0.4923 | 0.5033 | 0.4714 | 0.4095 |
| LSS | -0.0147 | 0.0000 | 0.1194 | 0.1211 | 0.0969 | -0.0144 |
| BSS | -0.0012 | 0.0000 | 0.0205 | 0.0208 | 0.0203 | 0.0124 |


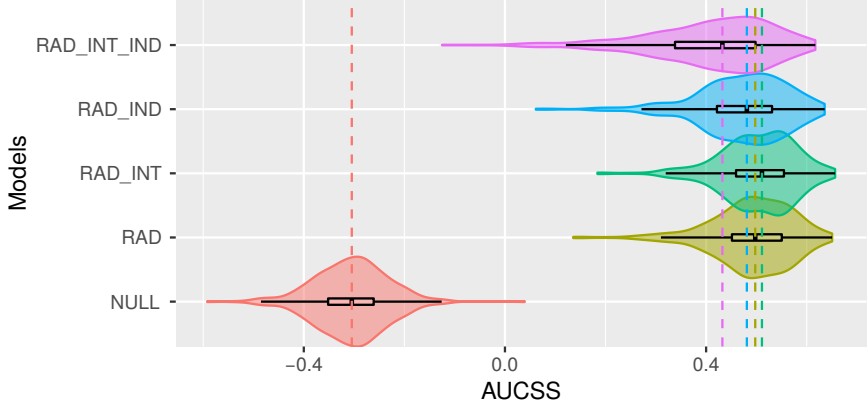

**Figure 1.** Distribution of cross-validated AUCSS values of 401 administrative districts is shown for different logistic regression models for weather-related accident probabilities. The probability density is smoothed by a kernel density estimator (shading). The median is indicated by vertical dashed lines.


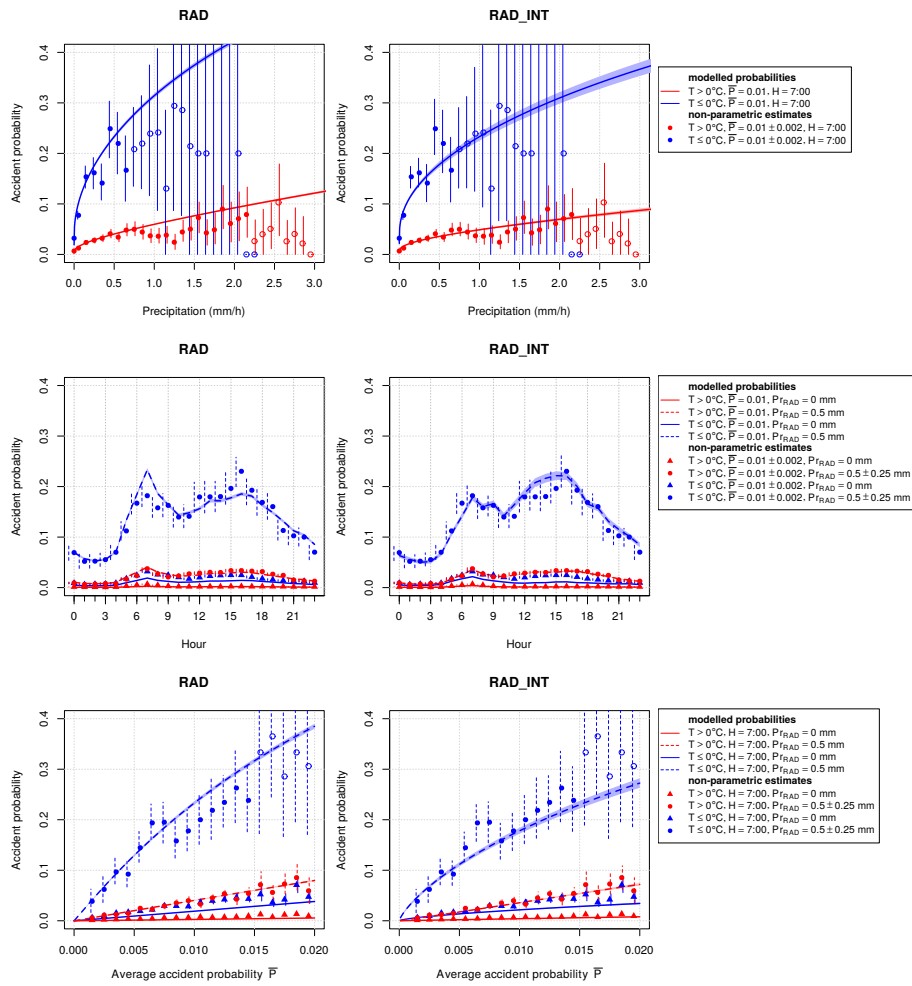

**Figure 2.** Comparison of modeled probabilities of weather-related road accidents with non-parametric probability estimates. Probabilities (lines) and 95% confidence intervals based on standard errors (shading) of model RAD (left) and RAD_INT (right) are displayed as a function of hourly precipitation (top), hour of the day (middle) and the temporal average accident probability of the administrative district (bottom) for different parameter settings (see legends for details). Non-parametric estimates of probabilities (markers) and 95% confident intervals based on bootstrapping (vertical lines) are shown for corresponding parameter ranges.

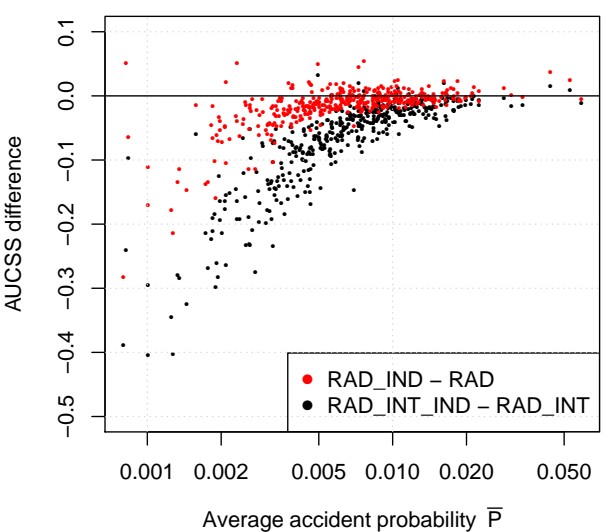

**Figure 3.** Differences of AUCSS values between the models RAD_IND and RAD (red) and RAD_INT_IND and RAD_INT (black). AUCSS differences are shown for each of the 401 administrative districts vs. the average accident probability $\overline{P}$ of the respective districts (dots).




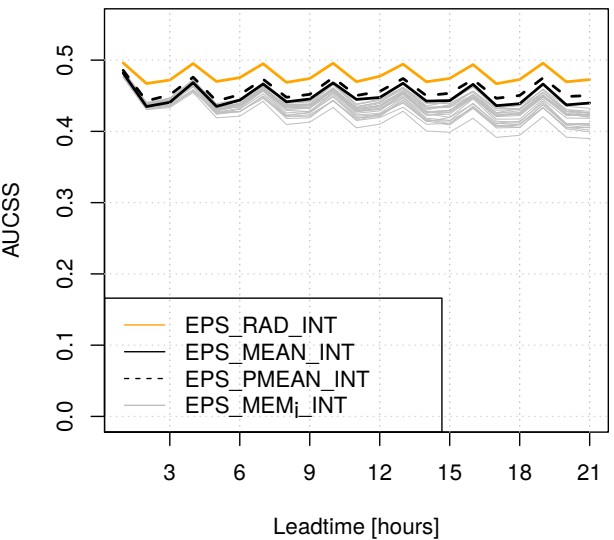

**Figure 4.** AUCSS values of different models for hourly probabilities of weather-related road accidents using radar, reanalysis and weather forecast data from 2011-2012 as a function of leadtime.

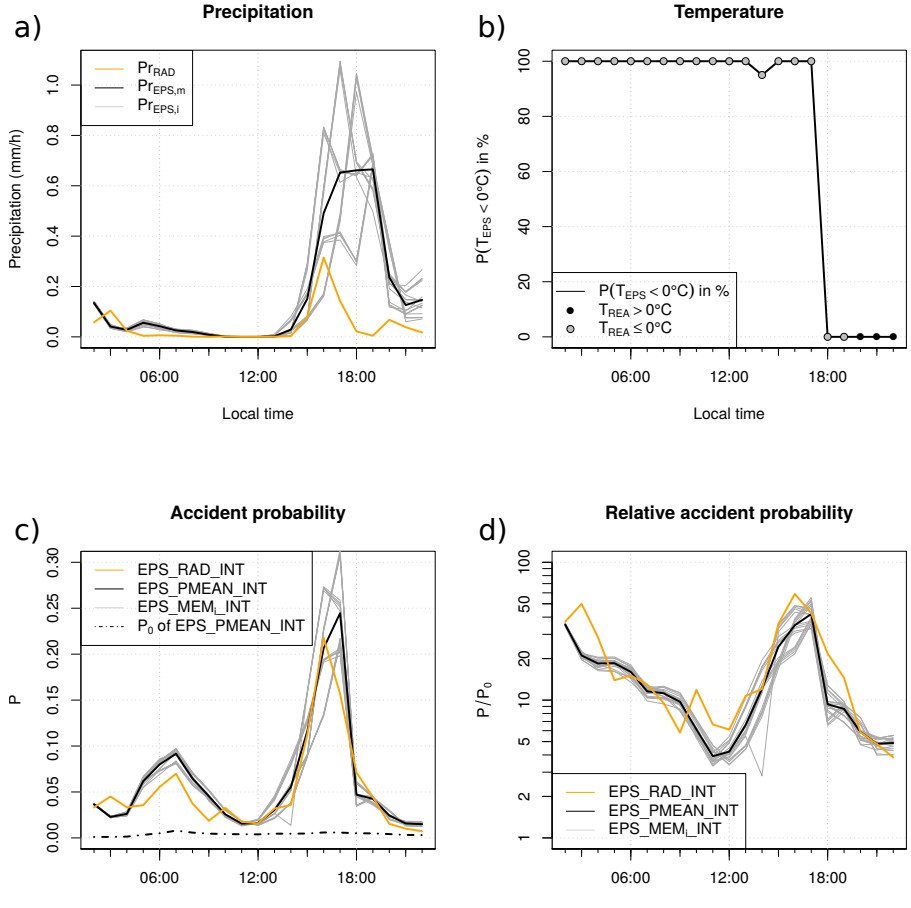

**Figure 5.** Application of the models EPS_RAD_INT and EPS_PMEAN_INT to a adverse winter weather event on 3rd Dec. 2012. Time series are shown for the district of Stuttgart using the COSMO-DE-EPS forecast initialized at 00 UTC. a) Hourly precipitation aggregated to district level, b) percentage of ensemble members with temperatures below 0°C, c) probability of weather-related road accidents and d) relative accident probability.


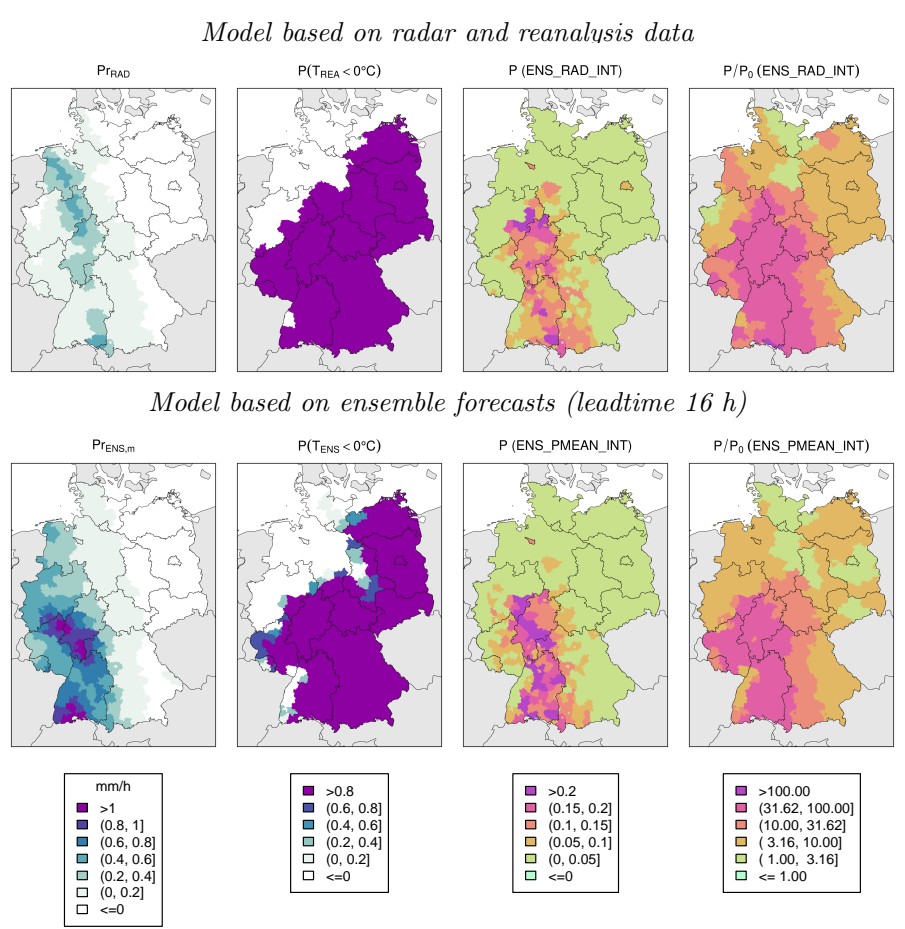

**Figure 6.** Model results for adverse winter weather conditions on 3rd Dec. 2012 at 17:00 local time based on models EPS_RAD_INT (top) and EPS_PMEAN_INT using the COSMO-DE-EPS forecast with a leadtime of 16 h initialized at 00 UTC (bottom). From left to right: hourly precipitation at district level, fraction of ensemble members with temperatures below 0°C, probability of weather-related road accidents, and relative accident probability.