# Peer review of "Predictive modeling of hourly probabilities for weather-related road accidents"

_Natural Hazards and Earth System Sciences, 2020_

## Referee Comment (RC1) · Anonymous Referee #1 · 27 Mar 2020

This paper is a sensitivity analysis of the most important factors which can lead to road collisions. The authors compare accident data to observed precipitation data, reanalysed data and ensemble weather forecasts then use logistical regression models to investigate which factors are most important when trying to forecast accidents in the future.

Overall the paper is interesting and well written. It uses appropriate data and the models are relevant. It is a useful paper for the scientific community. I recommend a few minor revisions before publication but otherwise happy for publication.

My comments are mainly suggestion for small edits to wording with details below: Abstract - Line 1 first word - use 'The' instead of 'An' - Line 2 - suggest 'This study investigates hourly...' instead of 'We study hourly...' - Line 8 - suggest 'approximately' instead

of 'about'

Introduction - Line 16 - space needed between 2016 and ( - Line 23 - add 'The' before 'aim' - Page 2, line 31 - check Mills et al reference - Page 3, line 9 - add 'The' before 'aim' - Page 3, line 16 and 17 - change 'Sect.' to 'Section'

Data - Page 4, line 4/5 - suggest 'Radar reflectives cannot...' instead of 'As from radar reflectives we cannot...' - Page 4, line 9 - suggest 'projects aim to combine the...' instead of 'projects thus aims at combining the...'

Methods - Page 5, line 15 - check brackets in equation - Page 5, line 29 - is a comma needed at end of equation? - Page 8, line 1/2 - suggest 'This allows the performance of the model for different districts to be assessed.'

Results - Page 9, line 4 - suggest 'P ranges from <0.001' instead of 'It ranges from below 0.001' - Page 9, line 25 - remove comma and include 'and' after 0 - Page 9, line 26 - clarify what 'they are' means - Page 9, line 32 - add 'a' after 'as'

Summary, discussion and conclusions - Page 13, line 24 - add 'by' after 'increases' - Page 13, line 25 - add 'the' after 'of' - Page 13, line 33 - add 'a' after 'that' - Page 14, line 3 - 'road user is rather interested in their individual...' instead of 'road used is rather interested in his individual...'

Tables and figures general comment - these should be able to stand on their own so acronyms need defining a much as possible. Table 2 - In caption refer to Table 1 for definitions of Formula variations Table 3 - de-acronym Figure 4 - can the 3 hour variations in the AUCSS be explained in the body of the text? Figure 6 - can the observed data be displayed to compare the model data to? This would be helpful to see to show that the models are a good representation and show which model set are better.

---

## Referee Comment (RC2) · Anonymous Referee #2 · 28 Mar 2020

The article investigates the effects of including weather-related information into a statistical model for predicting hourly probabilities of road accidents over Germany. The same analysis can be applied over different countries, provided that the required data sources are available.

This study fully exploits state-of-the-art meteorological data both in terms of high temporal and spatial resolution. Standard techniques are used for validation and verification. The contribution of this study to show the benefit of using meteorological data for predicting road accidents is valuable. As far as I can judge, there are no major flaws in the statistical analysis and the conclusions are well supported by the results. The presentation of the manuscript is clear and concise.

In conclusion, the study is valuable and worthy of publication.

[Figure]

Very minor comments:

- Table 2. RAD_INT. Please specify the meaning of the symbol P (it is not specified in Table 1)

- Page 2, line 30. "diving habits"

- Page 9, line 26. mm/h is in italic

---

## Referee Comment (RC3) · Anonymous Referee #3 · 14 May 2020

General comments

The paper deals with a very interesting subject, examining the impact of weather characteristics on hourly road accident probabilities, and assessing the respective models. It uses an appropriate methodology and produces promising novel research results. There are some issues in the present form of the paper that should be addressed before it is accepted for publication.

Specific comments

- On page 1, lines 17-18, it is mentioned that "weather is one of the most important factors contributing to road traffic safety". This is a strong statement that requires a corresponding reference. To the experience of the reviewer, there is a significant amount

of studies where weather-related variables are not as significant for road crashes as others (such as behavioral variables), or not at all.

- On page 2, lines 11-13, there is a very recent review on that point, and pertinent with the study in general, which the authors may want to consult:

Ziakopoulos, A., & Yannis, G. (2020). A review of spatial approaches in road safety. Accident Analysis & Prevention, 135, 105323.

- On page 3, lines 24-27, it is mentioned that "However, almost 8% of the accidents were indicated as being caused by adverse road conditions, which includes a wet, snowy or icy road, but also mud or dirt on the road. This class of accidents, which we refer to as weather-related accidents, is selected to generate the response variable used in the logistic regression models."

Firstly, it would be informative if the total number of considered accidents is mentioned (a rough calculation suggests it is about 345,000?). Secondly, and more importantly, this approach introduces a bias inherent from the subjectivity of crash recording, as it relies on indicators by policemen. The authors are suggested to elaborate on this bias, its extents and any implications it might have had on the results.

- For binary logistic models, the Hosmer-Lemeshow test is also customary to indicate the degree of correct predictions per population stratum. The authors can examine the HL for their best predictive models, or at least utilize it in future research.

- More importantly, a critical component of the study that is missing is a table with model coefficients (i.e. the influence of each variable) and their metrics (standard error, significance). The respective commentary of the effect of each variable is also critical. The authors should definitely add this part, at least for the best-performing models, as very useful knowledge and conclusions can be drawn, which are now left in the dark. After all, this is the main advantage of econometric models (such as logistic regression) vs. machine learning models, which are black boxes.

Technical corrections

- In the abstract, the authors mention 'skillful' predictions, which is an unclear term. Do they mean informed predictions? Furthermore, there is mention of model hit rates. Is this a percentage of accurate predictions? Please clarify these points so that the abstract is more comprehensive.

- On page 4, lines 25, it is stated that "$\tau$ is the difference between the time the model is initialized and the time the forecast is valid for". Shouldn't a more useful interval be between model finish and validity headway?

- The English language needs minor revisions throughout the paper and in the abstract to avoid typographical mistakes (e.g. assess instead of asses). Also the authors are urged to select either 'crash' (more widely used) or 'accident' and use a single term consistently throughout the text.

---

## Author Comment (AC1) · 25 Jun 2020

**Response to reviewer comments on "Predictive modeling of hourly probabilities for weather-related road accidents".**

N Becker, HW Rust, U Ulbrich

June 25, 2020

**Preliminaries**

We would like to thank the three anonymous reviewers for their comments on our manuscript. We find the comments helpful and constructive. We think that they will help to improve the manuscript.

In the following pages we set out in detail our responses to the comments and how we plan to act on them.

**Response to Anonymous Referee #1 (RC1)**

This paper is a sensitivity analysis of the most important factors which can lead to road collisions. The authors compare accident data to observed precipitation data, reanalysed data and ensemble weather forecasts then use logistical regression models to investigate which factors are most important when trying to forecast accidents in the future.

Overall the paper is interesting and well written. It uses appropriate data and the models are relevant. It is a useful paper for the scientific community. I recommend a few minor revisions before publication but otherwise happy for publication.

My comments are mainly suggestion for small edits to wording with details below:

**Abstract**

**Reviewer Comment C 1.1** — Line 1 first word - use 'The' instead of 'An'

**Reply**: We will use an alternative formulation using plural form.

**Reviewer Comment C 1.2** — Line 2 - suggest 'This study investigates hourly...' instead of 'We study hourly...'

**Reply**: We will follow the reviewers suggestion.

**Reviewer Comment C 1.3** — Line 8 - suggest 'approximately' instead of 'about'

**Reply**: We will follow the reviewers suggestion.

**Introduction**

**Reviewer Comment C 1.4** — Line 16 - space needed between 2016 and ( - Line 23 - add 'The' before 'aim'

**Reply**: We will correct the sentence following the reviewers suggestion.

**Reviewer Comment C 1.5** — Page 2, line 31 - check Mills et al reference

**Reply**: We will corrected the reference.

**Reviewer Comment C 1.6** — Page 3, line 9 - add 'The' before 'aim'

**Reply**: We will correct the sentence following the reviewers suggestion.

**Reviewer Comment C 1.7** — Page 3, line 16 and 17 - change 'Sect.' to 'Section'

**Reply**: We followed the abbreviation rules as described in the NHESS manuscript preparation guidelines for authors.

**Data**

**Reviewer Comment C 1.8** — Page 4, line 4/5 - suggest 'Radar reflectives cannot...' instead of 'As from radar reflectives we cannot...'

**Reply**: Radar reflectivity refers to the amount of radiation reflected back to the receiver by the precipitation particles

**Reviewer Comment C 1.9** — Page 4, line 9 - suggest 'projects aim to combine the...' instead of 'projects thus aims at combining the...'

**Reply**: We will follow the reviewers suggestion.

**Methods**

**Reviewer Comment C 1.10** — Page 5, line 15 - check brackets in equation

**Reply**: The used mathematical interval notation refers to a half-open interval. A half-open interval includes only one of its endpoints, and is denoted by mixing the notations for open and closed intervals. $(0, 1]$ means greater than 0 and less than or equal to 1, while $[0,1)$ means greater than or equal to 0 and less than 1.

**Reviewer Comment C 1.11** — Page 5, line 29 - is a comma needed at end of equation?

**Reply**: Yes, because the equation is part of the sentence.

**Reviewer Comment C 1.12** — Page 8, line 1/2 - suggest "This allows the performance of the model for different districts to be assessed."

**Reply**: We will change the sentence to "This allows us to compare the performance of the model in different districts.", because we want to emphasize that we are interested in the difference between the individual districts.

**Results**

**Reviewer Comment C 1.13** — Page 9, line 4 - suggest 'P ranges from ¡0.001' instead of 'It ranges from below 0.001'

**Reply**: We will reformulate the sentence to "It ranges from less than 0.001 ...".

**Reviewer Comment C 1.14** — Page 9, line 25 - remove comma and include 'and' after 0

**Reply**: We will correct the sentence following the reviewers suggestion.

**Reviewer Comment C 1.15** — Page 9, line 26 - clarify what 'they are' means

**Reply**: "The are" will be replaced by "Probabilities are".

**Reviewer Comment C 1.16** — Page 9, line 32 - add 'a' after 'as'

**Reply**: We correct the sentence following the reviewers suggestion.

**Summary, discussion and conclusions**

**Reviewer Comment C 1.17** — Page 13, line 24 - add 'by' after 'increases'

**Reply**: We will change the sentence to "We found that the probability of weather-related accidents depends on hourly precipitation to the power of 0.2." for clarity.

**Reviewer Comment C 1.18** — Page 13, line 25 - add 'the' after 'of'

**Reply**: We will correct the sentence following the reviewers suggestion.

**Reviewer Comment C 1.19** — Page 13, line 33 - add 'a' after 'that'

**Reply**: We will correct the sentence following the reviewers suggestion.

**Reviewer Comment C 1.20** — - Page 14, line 3 - 'road user is rather interested in their individual...' instead of 'road used is rather interested in his individual...'

**Reply**: We will correct the sentence following the reviewers suggestion.

**Tables and figures general comments**

**Reviewer Comment C 1.21** — these should be able to stand on their own so acronyms need defining a much as possible.

**Reply**: We will follow the reviewers suggestion and define all relevant acronyms in the figure and table captions.

**Reviewer Comment C 1.22** — Table 2 - In caption refer to Table 1 for definitions of Formula variations

**Reply**: We will follow the reviewers suggestion.

**Reviewer Comment C 1.23** — Table 3 - de-acronym

**Reply**: We will define the acronyms of the metrics displayed in table 3 as suggested.

**Reviewer Comment C 1.24** — Figure 4 - can the 3-hour variations in the AUCSS be explained in the body of the text?

**Reply**: The effect is explained in the second paragraph of section 4.2. The repetitive pattern occurs because hourly data is used for the analysis, but COSMO-DE-EPS is only initialized every three hours. Thus, the leadtimes 1, 4, 7, etc. include certain hours of the day, while the leadtimes 2, 5, 8, etc. include others. Consequently, there are three sets of lead times that are associated to different hours of the day, which causes differences in model performance for each set and leads to the observed three-hourly pattern.

**Reviewer Comment C 1.25** — Figure 6 - can the observed data be displayed to compare the model data to? This would be helpful to see to show that the models are a good representation and show which model set are better.

**Reply**: Unfortunately, the contractual obligations for the usage of the German accident data do not allow us to display information based on accident counts less than three to prevent the possibility of an identification of the drivers. Since in most districts one or two accidents occurred, the figure would be largely empty.

**Response to Anonymous Referee #2 (RC2)**

The article investigates the effects of including weather-related information into a statistical model for predicting hourly probabilities of road accidents over Germany. The same analysis can be applied over different countries, provided that the required data sources are available.

   This study fully exploits state-of-the-art meteorological data both in terms of high temporal and spatial resolution. Standard techniques are used for validation and verification. The contribution of this study to show the benefit of using meteorological data for predicting road accidents is valuable. As far as I can judge, there are no major flaws in the statistical analysis and the conclusions are well supported by the results. The presentation of the manuscript is clear and concise. In conclusion, the study is valuable and worthy of publication.

**Very minor comments**

**Reviewer Comment C 2.1** — Table 2. RAD_INT. Please specify the meaning of the symbol $P$ (it is not specified in Table 1)

**Reply**:  This was a mistake, we will replace $P$ with $\overline{P}'$.

**Reviewer Comment C 2.2** — Page 2, line 30. "diving habits"

**Reply**:  We will correct the sentence.

**Reviewer Comment C 2.3** — Page 9, line 26. mm/h is in italic

**Reply**:  We will change the unit to normal font type.

**Response to Anonymous Referee #3 (RC3)**

**General comments**

The paper deals with a very interesting subject, examining the impact of weather characteristics on hourly road accident probabilities, and assessing the respective models. It uses an appropriate methodology and produces promising novel research results. There are some issues in the present form of the paper that should be addressed before it is accepted for publication.

**Specific comments**

**Reviewer Comment C 3.1** — On page 1, lines 17-18, it is mentioned that "weather is one of the most important factors contributing to road traffic safety". This is a strong statement that requires a corresponding reference. To the experience of the reviewer, there is a significant amount of studies where weather-related variables are not as significant for road crashes as others (such as behavioral variables), or not at all.
On page 2, lines 11-13, there is a very recent review on that point, and pertinent with the study in general, which the authors may want to consult:
Ziakopoulos, A., & Yannis, G. (2020). A review of spatial approaches in road safety. Accident Analysis & Prevention, 135, 105323.

**Reply**: We agree with the reviewer that the statement that "weather is one of the most important factors contributing to road traffic safety" might be too strong in the given context. We will modify this part of the introduction. We also thank the reviewer for pointing us to the interesting review article of Ziakopoulos and Yannis (2020), which we will take into account in the revised version of the manuscript.

**Reviewer Comment C 3.2** — On page 3, lines 24-27, it is mentioned that "However, almost 8% of the accidents were indicated as being caused by adverse road conditions, which includes a wet, snowy or icy road, but also mud or dirt on the road. This class of accidents, which we refer to as weather-related accidents, is selected to generate the response variable used in the logistic regression models." Firstly, it would be informative if the total number of considered accidents is mentioned (a rough calculation suggests it is about 345,000?). Secondly, and more importantly, this approach introduces a bias inherent from the subjectivity of crash recording, as it relies on indicators by policemen. The authors are suggested to elaborate on this bias, its extents and any implications it might have had on the results.

**Reply**: Section 2.1 includes information about total accident numbers as well about numbers of time steps with at least one accident and their percentages. We will reformulate the section in a more consistent way. Also, we will correct some numbers given in that section, which have been taken over by mistake from a previous version of the manuscript. Therefore, they did not correspond to the data used in the present form of the study. Furthermore, we agree with the reviewer that a discussion of the subjectivity of the police officers decision on the accident cause is an important aspect. We will add a paragraph to the discussion section of the manuscript, where we address this issue.

**Reviewer Comment C 3.3** — For binary logistic models, the Hosmer-Lemeshow test is also customary to indicate the degree of correct predictions per population stratum. The authors can examine the HL for their best predictive models, or at least utilize it in future research.

**Reply**: The Hosmer-Lemeshow test (HL) is an interesting test we have not been aware of. It is comparable to the reliability component of the Brier score (BS) decomposition (Murphy, 1973). In both cases it is tested, whether or not the observed event rates match modeled event rates in certain subgroups of the modeled probabilities. In addition to the reliability, the BS decomposition includes a second component called "resolution". The resolution measures the distance between the observed relative frequency and climatological frequency. Thus, it indicates the degree to which the forecast can separate different situations. BS is a proper score which cannot be hedged (Wilks, 2011; Gneiting and Raftery, 2007; Jolliffe, 2008). HL instead cannot be proper as it can easily be hedged as the following example shows: A forecast always predicting the average probability is very reliable, but has a very low resolution, which is taken into account by the BS. The HL does not take resolution into account, but only tests for reliability. We tested the HL for our models and found that it is not suitable in our case. We find that our NULL model gets a perfect HL statistic of virtually 0, because it simply predicts the district average probabilities. The RAD_INT model, which includes meteorological predictor variables, gets a worse HL statistic and fails the significance test. We can assume that this corresponds to a reduction of reliability. However, since the HL does not take into account the resolution, it does not reward the RAD_INT model for "daring" to predict higher probabilities under adverse meteorological conditions. As suggested by the reviewer, we will consider the use of the HL in future studies, however, further research is necessary to test how the HL can be integrated into the concept of the BS decomposition for an improved consistency.

**Reviewer Comment C 3.4** — More importantly, a critical component of the study that is missing is a table with model coefficients (i.e. the influence of each variable) and their metrics (standard error, significance). The respective commentary of the effect of each variable is also critical. The authors should definitely add this part, at least for the best-performing models, as very useful knowledge and conclusions can be drawn, which are now left in the dark. After all, this is the main advantage of econometric models (such as logistic regression) vs. machine learning models, which are black boxes.

**Reply**: We agree with the reviewer that the model coefficients, standard errors and significances are important. However, since we use categorical variables and interaction terms the models in this study are relatively complex. For example, the best fitting model RAD_INT has 99 parameters, which are required to model the complex diurnal cycle based on 24 hourly coefficients and its interactions with the other parameters. Our idea was to base the description of the models in the results section of the article on the graphical representations in Figure 2, which are easier to read and interpret than a long table. Based on the reviewers comment, we decided to include the complete model coefficients, standard errors and significances of the models described in section 4.1 as supplementary material in the revised version of the paper and comment on that in the results section of the manuscript. We will provide the detailed model information in CSV format, which will enable the interested reader to look into the model details and easily reuse it for their own analyses. This will enhance the reproducibility of this study.

**Technical corrections**

**Reviewer Comment C 3.5** — - In the abstract, the authors mention 'skillful' predictions, which is an unclear term. Do they mean informed predictions? Furthermore, there is mention

of model hit rates. Is this a percentage of accurate predictions? Please clarify these points so that the abstract is more comprehensive.

**Reply**: We will reformulate the abstract to make it more comprehensive.

**Reviewer Comment C 3.6** — On page 4, lines 25, it is stated that "$\tau$ is the difference between the time the model is initialized and the time the forecast is valid for". Shouldn't a more useful interval be between model finish and validity headway?

**Reply**: For the verification of meteorological forecasts the leadtime $\tau$ is a standard parameter. It is used to assess how many hours/days ahead a forecast is useful. In contrast to a parameter that includes "model finish", as suggested by the reviewer, the leadtime *tau* is independent of the wall-clock-time, that actually passes from the start of the computer program to the end. We will add a sentence with an example to the manuscript to make the concept of leadtime more comprehensible for the reader.

**Reviewer Comment C 3.7** — The English language needs minor revisions throughout the paper and in the abstract to avoid typographical mistakes (e.g. assess instead of asses). Also the authors are urged to select either "crash" (more widely used) or "accident" and use a single term consistently throughout the text.

**Reply**: We will thoroughly check the manuscript for typographical mistakes and use the term "accident" consistently throughout the text.

**References**

T. Gneiting and A. E. Raftery. Strictly proper scoring rules, prediction, and estimation. *Journal of the American statistical Association*, 102(477):359–378, 2007.

I. T. Jolliffe. The impenetrable hedge: A note on propriety, equitability and consistency. *Meteorological Applications: A journal of forecasting, practical applications, training techniques and modelling*, 15(1):25–29, 2008.

A. H. Murphy. A new vector partition of the probability score. *Journal of applied Meteorology*, 12(4):595–600, 1973.

D. S. Wilks. *Statistical methods in the atmospheric sciences*, volume 100. Academic press, 2011.

A. Ziakopoulos and G. Yannis. A review of spatial approaches in road safety. *Accident Analysis & Prevention*, 135:105323, 2020.

---

## Author Response (AR1)

**Response to reviewer comments on "Predictive modeling of hourly probabilities for weather-related road accidents".**

N. Becker, H.W. Rust, U. Ulbrich

July 31, 2020

**Preliminaries**

We would like to thank the three anonymous reviewers for their comments on our manuscript. We find the comments helpful and constructive. We think that they helped to improve the manuscript.

In the following pages we set out in detail our responses to the comments and how we acted on them.

**Response to Anonymous Referee #1 (RC1)**

**Comment C 1.1** — Line 1 first word - use 'The' instead of 'An'

**Reply**: We used an alternative formulation using plural form.

**Comment C 1.2** — Line 2 - suggest 'This study investigates hourly...' instead of 'We study hourly...'

**Reply**: We followed the reviewers suggestion.

**Comment C 1.3** — Line 8 - suggest 'approximately' instead of 'about'

**Reply**: We followed the reviewers suggestion.

**Comment C 1.4** — Line 16 - space needed between 2016 and ( - Line 23 - add 'The' before 'aim'

**Reply**: We corrected the sentence following the reviewers suggestion.

**Comment C 1.5** — Page 2, line 31 - check Mills et al reference

**Reply**: We corrected the reference.

**Comment C 1.6** — Page 3, line 9 - add 'The' before 'aim'

**Reply**: We corrected the sentence following the reviewers suggestion.

**Comment C 1.7** — Page 3, line 16 and 17 - change 'Sect.' to 'Section'

**Reply**: We followed the abbreviation rules as described in the NHESS manuscript preparation guidelines for authors.

**Comment C 1.8** — Page 4, line 4/5 - suggest 'Radar reflectives cannot...' instead of 'As from radar reflectives we cannot...'

**Reply**: Radar reflectivity refers to the amount of radiation reflected back to the receiver by the precipitation particles

**Comment C 1.9** — Page 4, line 9 - suggest 'projects aim to combine the...' instead of 'projects thus aims at combining the...'

**Reply**: We followed the reviewers suggestion.

**Comment C 1.10** — Page 5, line 15 - check brackets in equation

**Reply**: The used mathematical interval notation refers to a half-open interval. A half-open interval includes only one of its endpoints, and is denoted by mixing the notations for open and closed intervals. $(0,1]$ means greater than 0 and less than or equal to 1, while $[0,1)$ means greater than or equal to 0 and less than 1.

**Comment C 1.11**  —  Page 5, line 29 - is a comma needed at end of equation?

**Reply**: Yes, because the equation is part of the sentence.

**Comment C 1.12**  —  Page 8, line 1/2 - suggest "This allows the performance of the model for different districts to be assessed."

**Reply**: We changed the sentence to "This allows us to compare the performance of the model in different districts.", because we want to emphasize that we are interested in the difference between the individual districts.

**Comment C 1.13**  —  Page 9, line 4 - suggest 'P ranges from <0.001' instead of 'It ranges from below 0.001'

**Reply**: We reformulated the sentence to "It ranges from less than 0.001 ...".

**Comment C 1.14**  —  Page 9, line 25 - remove comma and include 'and' after 0

**Reply**:  We corrected the sentence following the reviewers suggestion.

**Comment C 1.15**  —  Page 9, line 26 - clarify what 'they are' means

**Reply**:  "They are" was replaced by "Probabilities are".

**Comment C 1.16**  —  Page 9, line 32 - add 'a' after 'as'

**Reply**:  We corrected the sentence following the reviewers suggestion.

**Comment C 1.17**  —  Page 13, line 24 - add 'by' after 'increases'

**Reply**:  We changed the sentence to "We found that the probability of weather-related accidents depends on hourly precipitation to the power of 0.2." for clarity.

**Comment C 1.18**  —  Page 13, line 25 - add 'the' after 'of'

**Reply**:  We corrected the sentence following the reviewers suggestion.

**Comment C 1.19**  —  Page 13, line 33 - add 'a' after 'that'

**Reply**:  We corrected the sentence following the reviewers suggestion.

**Comment C 1.20**  —  - Page 14, line 3 - 'road user is rather interested in their individual...' instead of 'road used is rather interested in his individual...'

**Reply**:  We corrected the sentence following the reviewers suggestion.

**Tables and figures general comments**

**Comment C 1.21** — these should be able to stand on their own so acronyms need defining a much as possible.

**Reply**: We followed the reviewers suggestion and defined all relevant acronyms in the figure and table captions.

**Comment C 1.22** — Table 2 - In caption refer to Table 1 for definitions of Formula variations

**Reply**: We followed the reviewers suggestion.

**Comment C 1.23** — Table 3 - de-acronym

**Reply**: We defined the acronyms of the metrics displayed in table 3 as suggested.

**Comment C 1.24** — Figure 4 - can the 3-hour variations in the AUCSS be explained in the body of the text?

**Reply**: The effect is explained in the second paragraph of section 4.2. The repetitive pattern occurs because hourly data is used for the analysis, but COSMO-DE-EPS is only initialized every three hours. Thus, the lead times 1, 4, 7, etc. include certain hours of the day, while the lead times 2, 5, 8, etc. include others. Consequently, there are three sets of lead times that are associated to different hours of the day, which causes differences in model performance for each set and leads to the observed three-hourly pattern.

**Comment C 1.25** — Figure 6 - can the observed data be displayed to compare the model data to? This would be helpful to see to show that the models are a good representation and show which model set are better.

**Reply**: Unfortunately, the contractual obligations for the usage of the German accident data do not allow us to display information based on accident counts less than three to prevent the possibility of an identification of the drivers. Since in most districts one or two accidents occurred, the figure would be largely empty.

**Response to Anonymous Referee #2 (RC2)**

**Comment C 2.1** — Table 2. RAD_INT. Please specify the meaning of the symbol $P$ (it is not specified in Table 1)

**Reply**: This was a mistake, we replaced $P$ with $\overline{P}'$.

**Comment C 2.2** — Page 2, line 30. "diving habits"

**Reply**: We corrected the sentence.

**Comment C 2.3** — Page 9, line 26. mm/h is in italic

**Reply**: We changed the unit to normal font type.

**Response to Anonymous Referee #3 (RC3)**

**Comment C 3.1** — On page 1, lines 17-18, it is mentioned that "weather is one of the most important factors contributing to road traffic safety". This is a strong statement that requires a corresponding reference. To the experience of the reviewer, there is a significant amount of studies where weather-related variables are not as significant for road crashes as others (such as behavioral variables), or not at all.
On page 2, lines 11-13, there is a very recent review on that point, and pertinent with the study in general, which the authors may want to consult:
Ziakopoulos, A., & Yannis, G. (2020). A review of spatial approaches in road safety. Accident Analysis & Prevention, 135, 105323.

**Reply**: We agree with the reviewer that the statement that "weather is one of the most important factors contributing to road traffic safety" might be too strong in the given context. We modified this part of the introduction. We also thank the reviewer for pointing us to the interesting review article of Ziakopoulos and Yannis (2020), which we took into account on page 1 line 17 of the revised version of the manuscript.

**Comment C 3.2** — On page 3, lines 24-27, it is mentioned that "However, almost 8% of the accidents were indicated as being caused by adverse road conditions, which includes a wet, snowy or icy road, but also mud or dirt on the road. This class of accidents, which we refer to as weather-related accidents, is selected to generate the response variable used in the logistic regression models." Firstly, it would be informative if the total number of considered accidents is mentioned (a rough calculation suggests it is about 345,000?). Secondly, and more importantly, this approach introduces a bias inherent from the subjectivity of crash recording, as it relies on indicators by policemen. The authors are suggested to elaborate on this bias, its extents and any implications it might have had on the results.

**Reply**: Section 2.1 includes information about total accident numbers as well about numbers of time steps with at least one accident and their percentages. We reformulated the section in a more consistent way. Also, we corrected some numbers given in that section, which have been taken over by mistake from a previous version of the manuscript. Therefore, they did not correspond to the data used in the present form of the study. Furthermore, we agree with the reviewer that a discussion of the subjectivity of the police officers decision on the accident cause is an important aspect. We added a paragraph to the discussion section of the manuscript, where we address this issue.

**Comment C 3.3** — For binary logistic models, the Hosmer-Lemeshow test is also customary to indicate the degree of correct predictions per population stratum. The authors can examine the HL for their best predictive models, or at least utilize it in future research.

**Reply**: The Hosmer-Lemeshow test (HL) is an interesting test we have not been aware of. It is comparable to the reliability component of the Brier score (BS) decomposition (Murphy, 1973). In both cases it is tested, whether or not the observed event rates match modeled event rates in certain subgroups of the modeled probabilities. In addition to the reliability, the BS decomposition includes a second component called "resolution". The resolution measures the distance between the observed relative frequency and climatological frequency. Thus, it indicates the degree to which the forecast can separate different situations. BS is a proper

score which cannot be hedged (Wilks, 2011; Gneiting and Raftery, 2007; Jolliffe, 2008). HL instead cannot be proper as it can easily be hedged as the following example shows: A forecast always predicting the average probability is very reliable, but has a very low resolution, which is taken into account by the BS. The HL does not take resolution into account, but only tests for reliability. We tested the HL for our models and found that it is not suitable in our case. We find that our NULL model gets a perfect HL statistic of virtually 0, because it simply predicts the district average probabilities. The RAD_INT model, which includes meteorological predictor variables, gets a worse HL statistic and fails the significance test. We can assume that this corresponds to a reduction of reliability. However, since the HL does not take into account the resolution, it does not reward the RAD_INT model for "daring" to predict higher probabilities under adverse meteorological conditions. As suggested by the reviewer, we will consider the use of the HL in future studies, however, further research is necessary to test how the HL can be integrated into the concept of the BS decomposition for an improved consistency.

**Comment C 3.4** — More importantly, a critical component of the study that is missing is a table with model coefficients (i.e. the influence of each variable) and their metrics (standard error, significance). The respective commentary of the effect of each variable is also critical. The authors should definitely add this part, at least for the best-performing models, as very useful knowledge and conclusions can be drawn, which are now left in the dark. After all, this is the main advantage of econometric models (such as logistic regression) vs. machine learning models, which are black boxes.

**Reply**: We agree with the reviewer that the model coefficients, standard errors and significances are important. However, since we use categorical variables and interaction terms the models in this study are relatively complex. For example, the best fitting model RAD_INT has 99 parameters, which are required to model the complex diurnal cycle based on 24 hourly coefficients and its interactions with the other parameters. Our idea was to base the description of the models in the results section of the article on the graphical representations in Figure 2, which are easier to read and interpret than a long table. The effect of each variable on the accident probability is displayed and the standard errors are reflected by the confidence intervals. Based on the reviewers comment, we decided to include the complete model coefficients, standard errors and significances of the main models NULL, HOUR, RAD and RAD_INT, which are described in section 4.1, as supplementary material in the revised version of the paper and comment on that in the results section of the manuscript. We provide the detailed model information in CSV format, which will enable the interested reader to look into the model details and easily reuse it for their own analyses. This enhances the reproducibility of this study.

**Technical corrections**

**Comment C 3.5** — In the abstract, the authors mention "skillful" predictions, which is an unclear term. Do they mean informed predictions? Furthermore, there is mention of model hit rates. Is this a percentage of accurate predictions? Please clarify these points so that the abstract is more comprehensive.

**Reply**: With skillful we mean that the model has a positive skill score (see Eq. 5) and performs better than a reference model. We reformulated the abstract to make it more comprehensive.

**Comment C 3.6** — On page 4, lines 25, it is stated that "$\tau$ is the difference between the time the model is initialized and the time the forecast is valid for". Shouldn't a more useful interval be between model finish and validity headway?

**Reply**: For the verification of meteorological forecasts the lead time $\tau$ is a standard parameter. It is used to assess how many hours/days ahead a forecast is useful. In contrast to a parameter that includes "model finish", as suggested by the reviewer, the lead time $tau$ is independent of the wall-clock-time, that actually passes from the start of the computer program to the end. We added a sentence with an example to the manuscript to make the concept of lead time more comprehensible for the reader.

**Comment C 3.7** — The English language needs minor revisions throughout the paper and in the abstract to avoid typographical mistakes (e.g. assess instead of asses). Also the authors are urged to select either "crash" (more widely used) or "accident" and use a single term consistently throughout the text.

**Reply**: We thoroughly checked the manuscript for typographical mistakes and use the term "accident" consistently throughout the text.

[revised manuscript text omitted]